# Perioperative and Mid-Term Oncological and Functional Outcomes After Partial Nephrectomy for Entirely Endophytic Renal Tumors: A Prospective Multicenter Observational Study (The RECORD2 Project)

**DOI:** 10.3390/cancers17071236

**Published:** 2025-04-05

**Authors:** Fabrizio Di Maida, Andrea Mari, Daniele Amparore, Alessandro Antonelli, Riccardo Schiavina, Riccardo Giuseppe Bertolo, Alessandro Veccia, Eugenio Brunocilla, Riccardo Campi, Luigi Da Pozzo, Cristian Fiori, Paolo Gontero, Antonio Andrea Grosso, Luca Lambertini, Nicola Longo, Ciro Imbimbo, Alberto Briganti, Francesco Montorsi, Francesco Porpiglia, Luigi Schips, Nazareno Suardi, Sergio Serni, Bernardo Rocco, Andrea Minervini

**Affiliations:** 1Unit of Oncologic Minimally-Invasive Urology and Andrology, Department of Experimental and Clinical Medicine, University of Florence, Careggi Hospital, 50134 Florence, Italy; fabrizio.dimaida@unifi.it (F.D.M.); antonioandrea.grosso@unifi.it (A.A.G.); luca.lambertini@unifi.it (L.L.); andrea.minervini@unifi.it (A.M.); 2Division of Urology, Department of Oncology, San Luigi Gonzaga Hospital, School of Medicine, Orbassano, 10043 Turin, Italy; daniele.amparore@unito.it (D.A.); cristian.fiori@unito.it (C.F.); francesco.propiglia@unito.it (F.P.); 3Urology Clinic, Department of Surgery, Dentistry, Paediatrics and Gynaecology, University of Verona, Azienda Ospedaliera Universitaria Integrata di Verona, 37126 Verona, Italy; alessandro_antonelli@me.com (A.A.); riccardogiuseppe.bertolo@univr.it (R.G.B.); alessandro.veccia@univr.it (A.V.); 4Department of Urology, University of Bologna, 40126 Bologna, Italy; riccardo.schiavina3@unibo.it (R.S.); eugenio.brunocilla@unibo.it (E.B.); 5Department of Experimental, Diagnostic, and Specialty Medicine, University of Bologna, 40126 Bologna, Italy; sergio.serni@unifi.it; 6Unit of Urological Robotic Surgery and Renal Transplantation, Department of Experimental and Clinical Medicine, University of Florence, Careggi Hospital, 50134 Florence, Italy; riccardo.campi@unifi.it; 7Department of Urology, Papa Giovanni XXIII Hospital, 24127 Bergamo, Italy; ldapozzo@ospedaliriuniti.bergamo.it; 8Urology Unit, Department of Surgical Sciences, University of Turin, 10124 Turin, Italy; paolo.gontero@unito.it; 9Department of Urology, University Federico II of Naples, 80138 Naples, Italy; nicolalongo20@yahoo.it (N.L.); ciro.imbimbo@unina.it (C.I.); 10Unit of Urology, Division of Experimental Oncology, Urological Research Institute (URI), IRCCS San Raffaele Scientific Institute, 20132 Milan, Italy; briganti.alberto@hsr.it (A.B.); montorsi.francesco@hsr.it (F.M.); 11Department of Urology, SS Hospital. Annunziata, 66100 Chieti, Italy; luigi.schips@unich.it; 12Department of Urology, Ospedali Civili, University of Brescia, 25121 Brescia, Italy; nazareno.suardi@unibs.it; 13Department of Urology, ASST Santi Paolo e Carlo, 20146 Milan, Italy; bernardomariacesare.rocco@unicatt.it

**Keywords:** nephron-sparing surgery, endophytic tumor, partial nephrectomy, renal cell carcinoma, robotics

## Abstract

This study examines the outcomes of partial nephrectomy for entirely endophytic renal tumors, which are particularly challenging due to their deep location within the kidney. Using data from 211 patients treated at multiple Italian urological centers, the study compared open, laparoscopic, and robotic surgical approaches. The results indicate that while all three methods were effective, robotic partial nephrectomy was associated with lower blood loss, shorter surgery time, and better early kidney function preservation. However, the functional benefits diminished after two years. The study also found that older age and open surgery were independent predictors of worse kidney function recovery. Overall, partial nephrectomy for these tumors was found to be oncologically safe, with a recurrence-free survival rate of 93.8%. The findings suggest that robotic surgery should be preferred when feasible to optimize early kidney function recovery while maintaining good cancer control.

## 1. Introduction

Partial nephrectomy (PN) has become the gold standard for the treatment of small renal masses, particularly in patients with localized, stage T1 renal cell carcinoma (RCC) [1]. Its nephron-sparing advantage allows for the preservation of renal function while maintaining oncological efficacy comparable to that of radical nephrectomy (RN) [2,3]. RN, which involves the complete removal of the affected kidney, is typically reserved for larger or more complex renal tumors or when PN is not technically feasible. While RN ensures oncologic clearance, it carries a higher risk of postoperative chronic kidney disease, which has been associated with increased cardiovascular morbidity and mortality [4,5]. PN, in contrast, aims to excise the tumor with negative margins while minimizing the removal of healthy renal parenchyma. This surgical balance becomes particularly complex when dealing with tumors that are completely endophytic.

Totally endophytic tumors represent a unique subset of renal tumors, often associated with greater technical difficulty during PN [6,7]. These tumors are deeply embedded within the kidney, without protrusion into the perirenal fat or any part of the renal surface, making them difficult to localize, access, and excise. As a result, surgeons face challenges in obtaining adequate visualization, achieving negative surgical margins, and minimizing ischemia time. These tumors are often associated with higher surgical risks, including increased operative times, blood loss, and a potentially higher likelihood of conversion to RN [8,9].

Advancements in preoperative imaging and intraoperative guidance technologies have provided new opportunities to overcome the challenges posed by totally endophytic tumors. Innovations such as the 3D reconstruction [10] of preoperative imaging, intraoperative ultrasound [11,12], and robotic-assisted surgical platforms [13,14,15] have enhanced a surgeon’s ability to visualize and localize these tumors, allowing for more precise and less invasive resections. Robotic PN, in particular, has seen widespread adoption due to its advantages in terms of enhanced dexterity, improved visualization, and reduced recovery time for patients. Nonetheless, the treatment of totally endophytic renal tumors remains a highly complex and technically demanding procedure, and the literature on optimal strategies for managing these tumors through PN remains limited.

This manuscript seeks to explore the evolving role of PN in the management of totally endophytic renal tumors. Specifically, we sought to analyze data from our prospective, multi-institutional database aiming to assess functional and oncological outcomes of PN in patients with totally endophytic renal tumors treated with different surgical approaches. By doing so, this manuscript aims to provide surgeons with a deeper understanding of the complexities involved in treating totally endophytic tumors and to offer insights into optimizing nephron-sparing surgery in such a context.

## 2. Materials and Methods

### 2.1. Patient Selection and Dataset

The Italian Registry of Conservative and Radical Surgery for cortical renal tumor Disease (RECORD 2 Project) is a prospective observational multicenter project promoted by the Italian Society of Urology (SIU) [16]. This study was approved by the local ethics committee, and informed consent was collected for all the patients. Overall, 4308 consecutive patients who had PN and RN for cortical renal tumors at 26 urological Italian centers between 1st January 2013 and 3rd December 2016 were included. The patient selection flowchart from the RECORD2 project is depicted in Figure 1.

Main exclusion criteria included patients who underwent RN and those with exophytic or partially exophytic tumors. Additionally, patients with hereditary syndromes, or prior renal surgery on the same kidney, were also excluded to maintain a homogeneous study population. An online central data server was generated and centrally controlled to limit missing or wrong data inputs. All data were prospectively recorded by medical doctors. The database included 6 main folders: (1) anthropometric and preoperative data; (2) imaging, indications (elective, relative, and absolute), and co-morbidities; (3) intra-operative data; (4) post-operative data; (5) histological analysis; and (6) follow-up. Comorbidity status was evaluated by the Charslon comorbidity index (CCI) and the American Society of Anesthesiologists (ASA) physical status (PS) classification system. Performance status was evaluated using the Eastern Cooperative Oncology Group (ECOG) score. Surgical indications were defined as elective (localized unilateral RCC with healthy contralateral kidney), relative (localized unilateral RCC with the coexistence of comorbidities such as diabetes, hypertension, or lithiasis that could potentially affect kidney function in the future), and absolute (bilateral tumors, multiple tumors, moderate to severe chronic kidney disease, or tumors involving an anatomically or functionally solitary kidney). The choice between different surgical approaches was based on surgeon preferences. The Preoperative Aspects and Dimensions Used for an Anatomical (PADUA) score [17] was calculated to assess the nephrometric complexity of each case. Center caseload was defined as number of PNs per year. Renal function was measured as serum creatinine, and the estimated glomerular filtration rate (eGFR) was calculated using the Chronic Kidney Disease Epidemiology Collaboration (CKD-EPI) equation. A trifecta was defined as the contemporary presence of negative surgical margins, no perioperative complications, and ΔeGFR<25% at the 3rd postoperative day [18]. For the purpose of this study, only patients treated with PN for totally endophytic renal tumors were included. Figure 2 depicts the main steps of robotic PN for the surgical excision of entirely endophytic renal tumors. Patient follow-up was conducted in accordance with the European Association of Urology (EAU) guidelines for RCC [1]. Follow-up assessments included clinical evaluation, renal function monitoring (serum creatinine and eGFR), and radiological imaging at standardized intervals based on tumor stage and risk profile. These procedures ensured consistent and guideline-based surveillance across participating centers.

### 2.2. Statistical Analysis

For statistical purposes, independent variables included all patient- and tumor-related data available in our online prospectively maintained database. First, descriptive statistics were obtained by reporting medians (and interquartile ranges, IQR) for continuous variables, and frequencies and proportions for categorical variables, as appropriate. The probability of survival was assessed by the Kaplan-Meier method, with the log-rank test (Mantel-Cox) used to estimate differences among the levels of the analyzed variables. Multivariate logistic regression analysis was performed to evaluate predictors of RF loss >25% at 1 month follow up and Trifecta failure. The Receiver Operating Characteristic (ROC) curve was generated to discriminate between the predictive accuracy of selected variables for both outcomes. Statistical analyses were performed using STATA 16 (Stata Corp., College Station, TX, USA). All tests were two-sided with a significance set at *p* < 0.05.

## 3. Results

Overall, 211 patients were evaluated and 143 (67.8%) were male. Preoperative characteristics are reported in Table 1. Median PADUA score was 10 (IQR 9-11). An open, laparoscopic, and robotic approach was offered to 94 (44.5%), 52 (24.6%), and 65 (30.8%) patients, respectively. Patients treated with robotic PN were younger than those treated with open and laparoscopic PN (median age 60.1 years vs. 66 vs. 61.7, *p* = 0.04). No statistically significant difference was found in terms of tumor diameter and nephrometric characteristics in patients treated with different surgical approaches.

Intraoperative features are shown in Table 2. A retroperitoneal approach was preferred in 87.2% of open PNs, while a transperitoneal approach was chosen in 69.2% and 80% of laparoscopic and robotic PN (*p* = 0.001), respectively. A significantly higher percentage of pure enucleation was found in robotic PN, as compared to open and laparoscopic approaches (40% vs. 13.8% vs. 13.5%; *p* = 0.001). Patients submitted to robotic surgery were treated in centers with significantly higher caseload compared to open and laparoscopic approaches (71 vs. 62 vs. 49.5 PN/year; *p* = 0.04). The surgical approach did not affect intraoperative and postoperative complication rates, although robotic PNs were found to have significantly lower estimated blood loss (100 vs. 185 vs. 175 cc, *p* = 0.04) and shorter operative time (126 vs. 140 vs. 160; *p* = 0.01) as compared to open and laparoscopic PNs. The trifecta rate significantly differed among open, laparoscopic, and robotic PN groups (40.2% vs. 63.5% vs. 64.6%, respectively; *p* = 0.03).

Table 3 shows postoperative outcomes. Clear cell RCC, papillary RCC, and chromophobe RCC were found in 63.5%, 10.4%, and 7.1% of cases, while a benign histology was present in 14.2% of cases. Overall, 137 (64.9%), 35 (16.6%), and 9 (4.3%) renal tumors were staged as pT1a, pT1b, and pT3a, respectively. Positive surgical margins (PSMs) were reported in 18 (8.5%) patients, with no differences among different surgical approaches. At a median follow up of 36.3 (IQR 21.9−49.2) months, recurrence free survival was 93.8%. Cancer-specific survival was 100%. No differences in terms of survival outcomes emerged when comparing the three approaches. Kaplan–Meier curves, depicting RFS for open, laparoscopic, and robotic approaches, are shown in Figure 3.

The robotic approach was associated with a lower %eGFR drop at 1-month and 1-year evaluation. In particular, a significantly lower percentage of patients receiving robotic PN experienced a >25% eGFR drop at 1-month evaluation (15.4% vs. 42.6% vs. 32.7%; *p* = 0.01) and one-year assessment (12.3% vs. 25.5% vs. 23.1%; *p* = 0.01), with the benefit softening at 24 month-follow up (12.3% vs. 19.1% vs. 19.2%; *p* = 0.36) (Figure 4) (Table 4).

At multivariate analysis age at surgery (OR 1.05, 95%CI 1.01−1.11, and *p* = 0.04) and open surgical approach (OR:1.76, 95%CI 1.12−4.38, and *p* = 0.02) were confirmed as independent predictors for >25% renal function loss at 1-month follow-up. Similarly, age at surgery (OR 1.05, 95%CI 1.01−1.11, and *p* = 0.04) and open approach (OR 2.42, 95%CI 1.02−5.76, and *p* = 0.04) were found to be independently associated with the risk of Trifecta failure [Table 5]. To evaluate the performance of the multivariable logistic regression models, we generated ROC curves for two key outcomes (Figure 5). The model predicting Trifecta failure and 25% RF loss at 1 month showed areas under the curve (AUC) of 0.83 and 0.84, respectively.

## 4. Discussion

PN has proved to be a successful surgical option to wisely balance functional and oncological outcomes. In this context, endophytic tumors pose a significant challenge to surgeons due to the difficult visualization and excision of the renal mass, often leading to increasing technical difficulty.

Preoperative imaging plays a pivotal role in surgical planning. Advances in multiphase CT and MRI, as well as the integration of nephrometry scoring systems like PADUA and RENAL, have enabled better risk stratification and intraoperative decision-making. Furthermore, the use of intraoperative ultrasound [11,12], as highlighted in the literature, has become a cornerstone for guiding tumor enucleation and ensuring negative margins without the need for the excessive resection of healthy tissue.

Surgically, the evolution from open to minimally invasive and robotic-assisted approaches has redefined the technical limits of nephron-sparing surgery. Robotic PN, in particular, offers enhanced dexterity, 3D visualization, and ergonomic advantages, contributing to reduced warm ischemia time and improved perioperative outcomes. However, surgical experience and institutional volume remain critical determinants of success, as the learning curve for managing complex tumors is steep.

With advances rapidly evolving, robust data assessing oncological and functional outcomes for endophytic renal tumors represent a key unmet need. Moreover, the identification of the true determinants affecting RF drop after PN remains a highly controversial topic due to the non-negligible number of confounders that must be controlled for [19]. The results presented in this study aim at the better contextualization of the outcomes inherently associated with different approaches for the surgical excision of endophytic renal tumors. In particular, we sought to investigate whether the robotic approach may be beneficial in this scenario.

One key result is represented by an improved early functional recovery after robotic PN as compared to the laparoscopic and open approaches, with a much more noticeable difference at 1-month and 12-month follow-up, then significantly minimizing at 2-year assessment. This is a quite an intriguing finding, although also potentially influenced by contralateral kidney compensation occurring in a 24-month period. The protective role of robotic assistance over the RF drop after PN was further confirmed by our multivariate model. Looking at the current available literature, our data are consistent with those reported by Carbonara and coworkers in their multicenter comparative analysis [20]. Indeed, our one-year eGFR drop of 7.5 mL/min is aligned with their finding. The higher preservation of RF could be related to the maximal sparing of healthy parenchyma during tumor resection thanks to better visualization in robotic PN, a lower hypothetical parenchymal injury during the renorrhaphy phase, and lower bleeding. Interestingly enough, in our study the robotic approach was found to allow a higher percentage of the pure enucleation resection technique. In this regard, experienced surgeons acknowledge the fact that, in some cases of complex and totally endophytic renal tumors, tumor enucleation may be the only possible technique to forego major injuries and ultimately avoid RN [21]. Furthermore, surgeons may take advantage of the “enucleation” plane to safely and effectively excise even challenging masses, thus facilitating nephron-sparring renorrhaphy, which in turn likely influences postoperative functional outcomes [22,23].

The second key finding of the study was that age at surgery seems to be a predictive factor for both RF drop and Trifecta failure. This is consistent with several recent series reporting functional outcomes after robot-assisted PN [24] The correlation can be attributed to the natural loss of nephrons, both in quantity and quality, that occurs with aging, thereby decreasing the patient’s ability to recover from the surgical impact [25]. The study has successfully proved that patients with lower eGFR at baseline presented a proportionate risk of developing an RF drop after 1-month assessment. These results are in line with an increasing body of evidence underlining the crucial role of the quality of renal parenchyma at baseline in the prediction of the ultimate functional outcome after PN [26,27]. Using eGFR as a measure is advocated over relying solely on serum creatinine, as it has been shown that nearly a quarter of patients with normal serum creatinine levels still have an eGFR below 60 mL/min [28].

In terms of oncologic outcomes, PN is recognized for achieving an optimal balance between negative surgical margins and the preservation of renal function. Generally, the rate of PSMs following PN ranges between 5% and 10% [29]. This is also supported by Tellini and coworkers [30], who reported a PSM rate of 5.9% in their study including 459 patients undergoing PN via open retroperitoneal, laparoscopic, or robot-assisted approaches. In our study, PSMs were observed in 18 patients (8.5%), with no significant differences among the various surgical techniques. Specifically, the PSM rate after robotic PN was 7.8%, aligning closely with the 4.5% reported by Carbonara and coworkers [20]. More specifically, reported recurrence-free survival rates after PN for endophytic lesions typically exceed 90% at 3 to 5 years [31]. At a median follow-up of 36.3 months (IQR 21.9−49.2), our study demonstrated a noticeable recurrence-free survival rate of nearly 94%, thus confirming the optimal oncologic outcomes achieved by PN for complex tumor lesions, irrespective of the surgical approach chosen. However, it is important to interpret these results with caution. The primary focus of this analysis was on perioperative and functional outcomes rather than oncologic endpoints, and the study was not specifically powered to detect subtle differences in recurrence or survival between surgical approaches. Moreover, the cohort largely consisted of patients with low-stage disease (e.g., pT1a), which may have contributed to the favorable survival outcomes observed across all groups.

The present study is not devoid of several limitations. The lack of randomization in the choice of the surgical approach to PN may have introduced statistical biases. We also noticed that a discrepancy between volume caseloads exists among different hospitals enrolling in the study; hence, several findings may have been influenced by different surgical experiences. Acknowledging these limitations, the present study represents one of the largest series so far exploring oncologic and functional results after PN performed with different surgical approaches for totally endophytic renal tumors. Larger randomized clinical studies are eagerly warranted to corroborate our findings.

## 5. Conclusions

PN is a feasible and effective treatment option for entirely endophytic renal tumors, providing excellent perioperative, functional, and oncologic outcomes across open, laparoscopic, and robotic approaches. Robotic surgery was associated with better early renal function preservation, while long-term outcomes were similar among techniques.

Favorable recurrence-free and cancer-specific survival rates support the oncologic safety of nephron-sparing surgery in these technically complex cases. These results highlight the importance of surgical expertise, careful preoperative planning, and interdisciplinary collaboration.

Ongoing research should further explore long-term functional outcomes, patient-centered metrics, and the role of advanced imaging in refining treatment strategies.

## Figures and Tables

**Figure 1 cancers-17-01236-f001:**
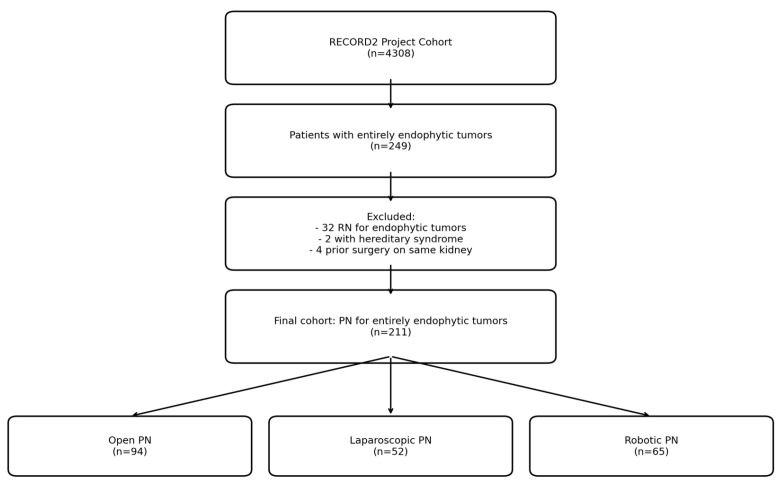
Patient selection flowchart from the RECORD2 project.

**Figure 2 cancers-17-01236-f002:**
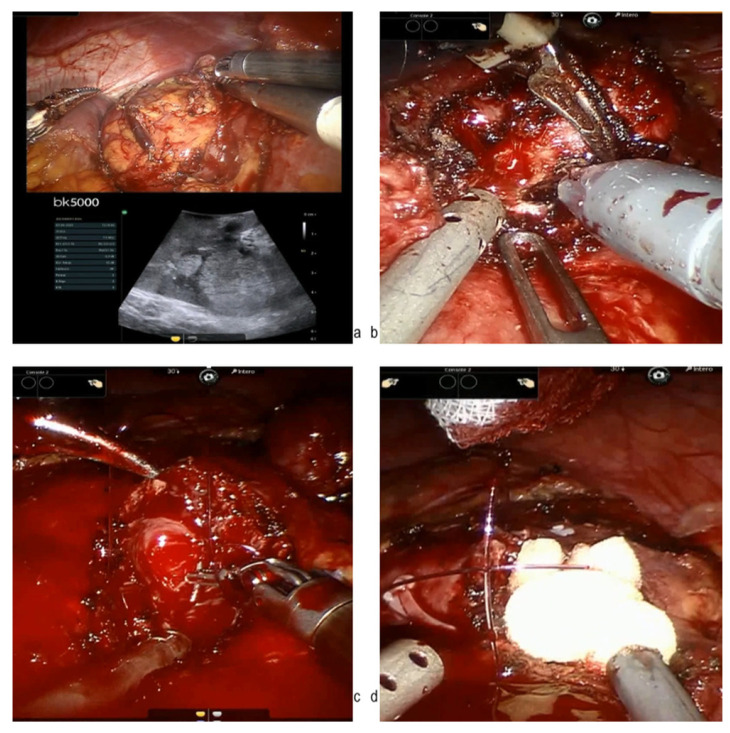
Surgical steps of robotic PN for entirely endophytic renal tumors. (**a**) Intraoperative ultrasound allows one to mark the enucleation margins; (**b**) pure enucleation for an entirely endophytic renal tumor; (**c**) double-layer renorrhaphy; and (**d**) hemostatic agent placement.

**Figure 3 cancers-17-01236-f003:**
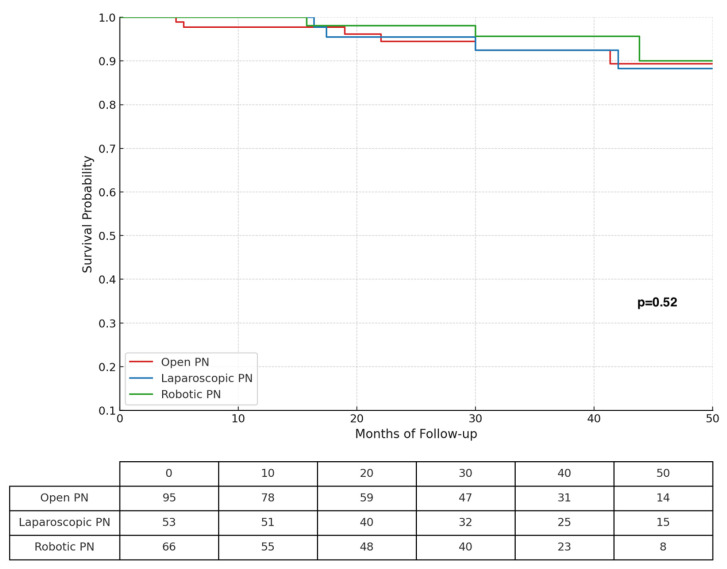
Kaplan–Meier curves depicting recurrence-free survival according to different surgical approaches.

**Figure 4 cancers-17-01236-f004:**
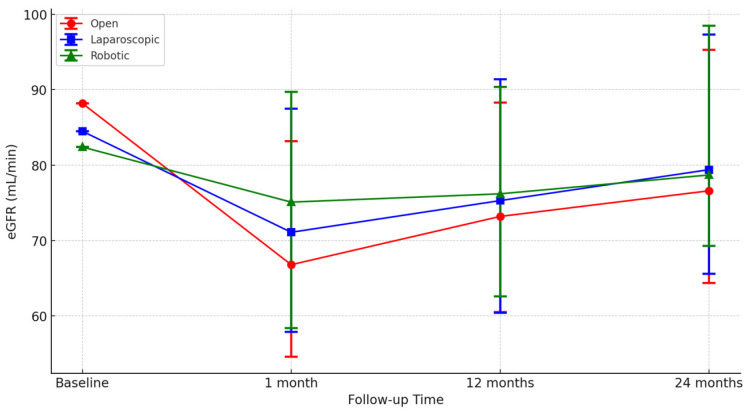
Median estimated glomerular filtration rate (eGFR) and interquartile ranges over time for each surgical approach—open, laparoscopic, and robotic PN.

**Figure 5 cancers-17-01236-f005:**
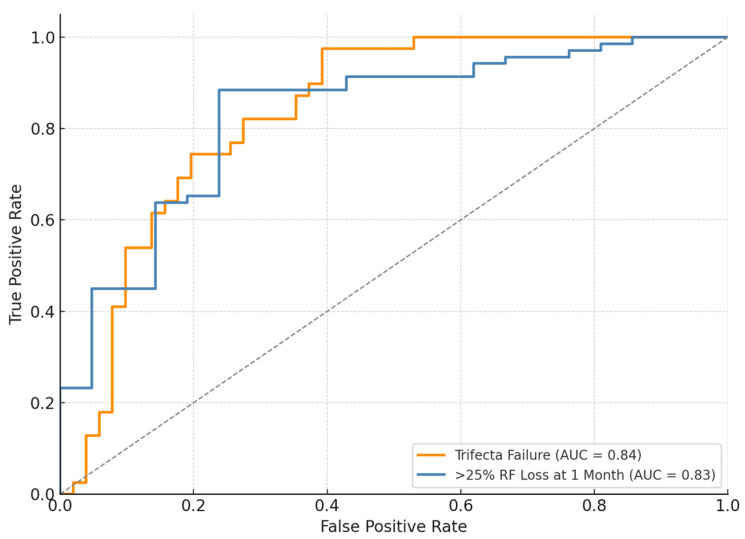
ROC curves from multivariable logistic regression models predicting two postoperative outcomes following partial nephrectomy for entirely endophytic renal tumors. The orange curve represents the model predicting Trifecta failure, while the blue curve represents the model predicting >25% renal function loss at 1 month. The area under the curve (AUC) reflects the model’s discriminative ability. Dashed diagonal line indicates reference (random) performance.

**Table 1 cancers-17-01236-t001:** Baseline features of 211 patients treated with partial nephrectomy for entirely endophytic renal tumors.

Preoperative Features		Total (n = 211)	Open (n = 94)	Laparoscopic (n = 52)	Robotic (n = 65)	*p* Value
Age (years), median IQR		63.6 (52.9−70.9)	66 (54.8−72.6)	61.7 (49.5−67.4)	60.1 (51.6−68.9)	**0.04**
Gender, n%	Male	143 (67.8)	60 (63.8)	40 (76.9)	43 (66.2)	0.26
	Female	68 (32.2)	34 (36.2)	12 (23.1)	22 (33.8)	
BMI (kg/m^2^), median IQR		25.8 (23.7−28.7)	25.5 (23.5−28.4)	26.1 (24.0−28.2)	26.1 (23.9−28.9)	0.87
ECOG Score	median IQR	0 (0−0)	0 (0−1)	0 (0−0)	0 (0−1)	0.16
	≥ 1, n %	49 (23.2)	17 (18.1)	12 (23.1)	20 (30.7)	0.19
ASA Score	median IQR	2 (2−3)	2 (2−3)	2 (2−3)	2 (2−2)	**0.01**
	≥ 3, n %	53 (25.1)	27 (28.7)	15 (28.8)	11 (16.9)	0.41
CCI PS score, median IQR		0 (0−0)	0 (0−2)	0 (0−1)	0 (0−2)	0.52
AA-CCI PS score, median IQR			4 (2−5)	3 (2−4)	3 (2−4)	0.05
Surgical indication, n %	Elective	187 (88.9)	83 (88.3)	47 (90.4)	57 (87.7)	0.89
	Relative	19 (9.0)	7 (7.4)	4 (7.7)	8 (12.3)	
	Imperative	5 (2.4)	4 (4.3)	1 (1.9)	0 (0.0)	
Tumor side, n. %	Right	122 (57.8)	57 (60.6)	28 (53.8)	37 (56.9)	0.74
	Left	86 (40.8)	35 (37.2)	23 (44.2)	28 (43.1)	
	Bilateral	3 (1.4)	2 (2.1)	1 (1.9)	0 (0.0)	
Clinical T, n. %	T1a	178 (84.4)	81(86.2)	43 (82.7)	54 (83.1)	0.82
	T1b	29 (13.7)	11 (11.7)	8 (15.4)	10 (15.4)	
	T2a	1 (0.5)	1 (1.1)	0 (0.0)	0 (0.0)	
	T3a	3 (1.4)	1 (1.1)	1 (1.9)	1 (1.5)	
Multiple ipsilateral lesion, n. %		10 (4.7)	6 (6.4)	1 (1.9)	3 (4.6)	0.48
Renal margin, n. %	Lateral	100 (47.4)	43 (45.7)	27 (51.9)	30 (46.2)	0.75
	Medial	111 (52.6)	51 (54.3)	25 (48.1)	35 (53.8)	
Renal face, n. %	Anterior	115 (54.5)	54 (57.4)	26 (50.0)	35 (53.8)	0.68
	Posterior	96 (45.5)	40 (42.6)	26 (50.0)	30 (46.2)	
Tumor location relative to the polar line (PL), n %	Entirely above PL	63 (29.9)	20 (21.3)	20 (38.5)	23 (35.4)	0.35
	≤ 50% crosses PL	83 (39.3)	45 (47.9)	16 (30.8)	22 (33.8)	
	> 50% crosses PL	65 (30.8)	29 (30.9)	16 (30.8)	20 (30.8)	
PADUA score, median IQR		10 (9−11)	10 (9−11)	9 (9−11)	10 (9−11)	0.24
PADUA risk classes, n %	Low risk	0 (0.0)	0 (0.0)	0 (0.0)	0 (0.0)	0.21
	Intermediate risk	85 (40.3)	33 (35.1)	26 (50.0)	26 (40.0)	
	High risk	126 (59.7)	61 (64.9)	26 (50.0)	39 (60.0)	
RENAL score, median IQR		9 (8−10)	9 (8−10)	8 (7−9)	8 (7−10)	0.10
Preoperative biopsy, n %		7 (3.3)	3 (3.2)	1 (1.9)	3 (4.6)	0.72
Baseline hemoglobin (mg/dL), median (IQR)		14.3 (13.4−15.1)	14.3 (13.3−15.1)	14.5 (13.4−15.1)	14.3 (13.4−15.3)	0.74
Baseline creatinine (mg/dL), median (IQR)		0.9 (0.8−1.1)	0.9 (0.7−1.0)	0.95 (0.8−1.1)	0.9 (0.8−1.1)	0.06
Baseline eGFR (mL/min), median IQR		85.1 (71.9−100.7)	88.2 (74.2−103.4)	84.5 (69.6−101.1)	82.4 (72.4−95.8)	0.45

**Table 2 cancers-17-01236-t002:** Intraoperative features of 211 patients treated with partial nephrectomy for entirely endophytic renal tumors.

Surgical Features		Open (n = 94)	Laparoscopic (n = 52)	Robotic (n = 65)	*p* Value
Approach, n. %	Transperitoneal	12 (12.8)	36 (69.2)	52 (80.0)	**0.001**
	Retroperitoneal	82 (87.2)	16 (30.8)	13 (20.0)	
Type of resection, n. %	Pure Enucleation	13 (13.8)	7 (13.5)	26 (40.0)	**0.001**
	Standard PN	81 (86.2)	45 (86.5)	39 (60.0)	
Hilar clamping, n. %	Performed	57 (60.6)	29 (55.8)	45 (69.2)	0.31
	Not performed	37 (39.4)	23 (44.2)	20 (30.8)	
Warm ischemia time (min), median IQR		17 (13−23)	15 (12.5−22.5)	18 (13−24)	0.74
Volume center (number of PN/year in each center), median IQR		62 (37−75)	49.5 (32−79)	71 (43.5−118)	**0.04**
Patients treated in centers performing > 30 PN/year, n. %		75 (79.8)	42 (80.8)	50 (76.9)	0.86
Patients treated in centers performing > 50 PN/year, n. %		59 (62.8)	32 (61.5)	51 (78.5)	0.07
EBL (cc), median IQR		185 (87.5−362.5)	175 (57−475)	100 (60−200)	**0.04**
Operative time (minutes), median IQR		140 (110−170)	160 (132−215)	126 (90−175)	**0.01**
Hemostatic agents, n. %		27 (28.7)	4 (7.7)	16 (24.6)	**0.01**
Renorrhaphy, n. %		90 (95.7)	50 (96.2)	55 (84.6)	**0.02**
Intraoperative surgical complications, n. %		3 (3.2)	3 (5.8)	2 (3.1)	0.69
Conversion rate, n. %		0 (0.0)	1 (1.9)	0 (0.0)	0.22
Postoperative surgical complications, n. (%)		15 (16.0)	4 (7.7)	5 (7.7)	0.06
○ Clavien Dindo ≥ 3, n. (%)		2 (2.1)	1 (1.9)	1 (1.5)	0.54
Medical complications, n. (%)		4 (4.2)	2 (3.8)	2 (3.1)	0.36
Time do drainage removal (days), median (IQR)		4 (3−5)	3 (2−4)	3 (2−4)	**0.04**
Length of stay (days), median (IQR)		5 (5−6)	3 (3−4)	3 (3−4)	**0.01**
Trifecta rate, n. %		38 (40.2)	33 (63.5)	42 (64.6)	**0.03**

**Table 3 cancers-17-01236-t003:** Postoperative and pathologic features of 211 patients treated with partial nephrectomy for entirely endophytic renal tumors.

Pathological Features		Open (n = 94)	Laparoscopic (n = 52)	Robotic (n = 65)	*p* Value
Pathological diameter, median (IQR)		3 (2−4)	3 (2−4)	3 (2−3)	0.39
Tumor histotype, n. %	• Clear cell RCC	63 (67.0)	28 (53.8)	43 (66.2)	0.95
	• Papillary RCC	7 (7.4)	7 (13.5)	8 (12.3)	
	• Chromophobe RCC	4 (4.3)	8 (15.4)	3 (4.6)	
	• Oncocytoma	13 (13.8)	4 (7.7)	10 (15.4)	
	• Angiomiolypoma	2 (2.1)	1 (1.9)	0 (0.0)	
	• Other	5 (5.3)	4 (7.7)	1 (1.5)	
Fuhrman grade, n. %	• Grade 1	6 (6.4)	3 (5.8)	4 (6.2)	0.96
	• Grade 2	47 (50.0)	26 (50.0)	35 (53.8)	
	• Grade 3	17 (18.1)	9 (17.3)	13 (20.0)	
	• Grade 4	1 (1.1)	1 (1.9)	0 (0.0)	
	• Absent/not applicable	23 (24.5)	13 (25.0)	13 (20.0)	
pT stage, n. %	• pT1a	62 (66.0)	33 (63.5)	42 (64.6)	0.62
	• pT1b	17 (18.1)	8 (15.4)	10 (15.4)	
	• pT3a	1 (1.1)	5 (9.6)	3 (4.6)	
	• Absent/not applicable	14 (14.9)	6 (11.5)	10 (15.4)	
Positive surgical margins, n. %		7 (7.4)	6 (11.5)	5 (7.8)	0.62
Urinary calyceal system invasion, n. %		0 (0.0)	2 (3.8)	3 (4.6)	0.12
Lymphovascular invasion, n. %		4 (4.3)	4 (7.7)	1 (1.5)	0.26
Tumor necrosis, n. %		10 (10.6)	9 (17.3)	6 (9.2)	0.36

**Table 4 cancers-17-01236-t004:** Functional and oncologic features of 211 patients treated with partial nephrectomy for entirely endophytic renal tumors.

Functional Outcomes	Open (n = 94)	Laparoscopic (n = 52)	Robotic (n = 65)	*p* Value
eGFR at 1 month follow up, median (IQR)	66.8 (54.6−83.2)	71.1 (57.9−87.5)	75.1 (58.4−89.7)	**0.04**
Preop—1 month follow up Δ eGFR, (%) (IQR)	24.3 (9.4−32.2)	15.9 (7.4−22.3)	8.9 (2.3−14.2)	**0.01**
>25% RF loss at 1 month follow up, n (%)	40 (42.6)	17 (32.7)	10 (15.4)	**0.01**
eGFR at 12 months follow up, median (IQR)	73.2 (60.5−88.3)	75.3 (60.4−91.4)	76.2 (62.6−90.4)	0.28
Preop—12 months follow up Δ eGFR, (%) (IQR)	17 (5.4−29.8)	10.9 (3.9−17.4)	7.5 (2.4−15.6)	**0.03**
>25% RF loss at 12 months follow up, n (%)	24 (25.5)	12 (23.1)	8 (12.3)	**0.01**
eGFR at 24 months follow up, median (IQR)	76.6 (64.4−95.3)	79.4 (65.6−97.3)	78.7 (69.3−98.5)	0.42

**Table 5 cancers-17-01236-t005:** Multivariate analysis exploring predictors of >25% renal function loss at 1-month follow-up and Trifecta failure.

		RF Loss > 25% at 1 Month Follow Up			Trifecta Failure		
Covariates		OR	95%CI	*p* value	OR	95%CI	*p* value
Age at surgery		1.05	1.01−1.11	**0.04**	1.05	1.01−1.11	**0.04**
Gender	Male	0.78	0.35−1.74	0.54	0.89	0.41−1.95	0.79
	Female (ref)	-	-	-	-	-	-
Charlson comorbidity index		0.99	0.74−1.33	0.97	1.02	0.69−1.21	0.54
Preoperative eGFR		1.03	1.01−1.06	**0.01**	1.01	0.99−1.04	0.21
PADUA score		1.18	0.86−1.62	0.31	1.16	0.86−1.56	0.33
WIT		1.02	0.96−1.06	0.63	1.01	0.96−1.06	0.67
Surgical approach	Open	1.76	1.12−4.38	**0.02**	2.42	1.02−5.76	**0.04**
	Laparoscopic	1.12	0.33−2.86	0.23	1.87	0.69−5.05	0.22
	Robotic (ref)	-	-	-	-	-	-

## Data Availability

The data presented in this study are available on request from the corresponding author due to privacy policy.

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
