# Peer review of "Perioperative and Mid-Term Oncological and Functional Outcomes After Partial Nephrectomy for Entirely Endophytic Renal Tumors: A Prospective Multicenter Observational Study (The RECORD2 Project)"

_cancers, 2025, doi:10.3390/cancers17071236_

Round 1

Reviewer 1 Report

Comments and Suggestions for Authors

Please find my comments:
What specific follow-up was applied? EAU guidelines?

Why did only 3% of patients undergo a preoperative biopsy?

What histopathological types were diagnosed in the other cases? (Line 166 and Table 3)

I don’t think it’s a good idea to analyze recurrence-free survival and cancer-specific survival independently from TNM classification, histopathological type, and grade.

Please consider adding some radiological images or surgical pictures to better illustrate the topic.

Please consider adding a GFR graph for better visualization of the topic.

Author Response

  • What specific follow-up was applied? EAU guidelines?

We thank the Reviewer for his/her valuable comment. The patient follow-up schedule was based on the EAU recommendations, with regular intervals as per guideline specifications. All follow-up procedures were documented in accordance with EAU standards to ensure consistency and comparability with guideline-based practices. We have clarified this issue in Materials and Methods section

  • Why did only 3% of patients undergo a preoperative biopsy?

Thank you for your insightful question. The low percentage of patients undergoing preoperative biopsy (3%) reflects the fact that this procedure is not routinely performed in most centers when it does not change clinical management. In daily practice, biopsy is generally reserved for cases where it could alter therapeutic decisions. However, for most renal masses, surgical intervention is guided by imaging and clinical assessment rather than histological confirmation.

Moreover, preoperative biopsy has notable limitations. It may be insufficient due to intratumoral heterogeneity, where a single sample does not necessarily represent the entire tumor composition. Additionally, it carries potential complications, including bleeding, infection, and, although rare, the risk of tumor seeding. Given these factors, most centers do not routinely apply preoperative biopsy as part of their standard protocol.

It is important to highlight that this study was based on an Italian registry, meaning that the decision to perform a preoperative biopsy was left to the discretion of individual clinicians. Physicians had the autonomy to proceed with a biopsy whenever they deemed it necessary, which aligns with the real-world variability in clinical practice. The low percentage observed in our study, therefore, reflects standard clinical decision-making rather than a methodological limitation. 

  • What histopathological types were diagnosed in the other cases? (Line 166 and Table 3)

Thank you for your question. The histopathological types diagnosed in the "other" cases, as mentioned in Line 166 and Table 3, included a variety of less common renal tumor subtypes that did not fall under the main histological categories presented separately. These cases comprised rare histological variants like metanephric adenoma, renal medullary carcinoma, multilocular cystic renal neoplasm of low malignant potential and tubulocystic renal cell carcinoma. Given the relatively small number of these cases, they were grouped together under the "other" category to maintain clarity and statistical relevance in the analysis.

  • I don’t think it’s a good idea to analyze recurrence-free survival and cancer-specific survival independently from TNM classification, histopathological type, and grade.

We thank the Reviewer for his/her valuable comment. We agree that analyzing recurrence-free survival (RFS) and cancer-specific survival (CSS) without stratifying or adjusting for TNM classification, histopathological subtype, and tumor grade could limit the interpretability and clinical relevance of the findings. These factors are well-established prognostic determinants in renal cell carcinoma and significantly influence both oncologic and functional outcomes.

In the RECORD2 study, although RFS was reported as high (93.8%) and CSS reached 100% across surgical approaches, it's important to note that the cohort primarily included low-stage tumors (e.g., pT1a in 64.9% of cases), and clear cell RCC was the predominant histological type (63.5%). Without multivariable stratification or subgroup analysis based on TNM stage, grade (e.g., Fuhrman), and histopathological type, the oncological outcomes might be overly optimistic or not generalizable to higher-risk or more aggressive tumor profiles.

Furthermore, multivariate analysis was used primarily to assess predictors of renal function loss and Trifecta failure, but not oncologic outcomes like recurrence or cancer-specific death. This omission underscores the risk of confounding—patients with more aggressive histology or higher-grade tumors may be underrepresented or have received different management, thus biasing survival estimates.

For a more robust understanding of oncologic safety, future analyses should incorporate these pathological factors to provide stratified survival outcomes and better inform surgical decision-making.

  • Please consider adding some radiological images or surgical pictures to better illustrate the topic.

We thank the Reviewer for his/her valuable comment. We have added some surgical pictures to better highlight surgical phases of tumor excision in this particular setting of tumors.

  • Please consider adding a GFR graph for better visualization of the topic.

Thank you for the valuable suggestion. In response, we have included a graph illustrating the median estimated glomerular filtration rate (eGFR) trends over time for the three surgical approaches—open, laparoscopic, and robotic partial nephrectomy. This visual representation enhances the understanding of renal function outcomes postoperatively and supports the data presented in the manuscript. We believe this addition improves the overall clarity and impact of the findings.

Reviewer 2 Report

Comments and Suggestions for Authors

The article Perioperative and Mid-Term Oncological and Functional Outcomes After Partial Nephrectomy for Entirely Endophytic Renal Tumors: A Prospective Multicenter Observational Study (The RECORD2 Project) discusses interesting oncological information. Suggestions:

  1. The introduction is too brief. Provide more information from the literature, explaining in detail what nephrectomy (both total and partial) involves. Also, discuss how modern surgical techniques, including the use of ultrasound, have reached new limits – 10.3390/diagnostics14090942.
  2. Add a flowchart in Chapter 2.
  3. Provide more detailed information about the exclusion criteria.
  4. Mark statistical significances clearly.
  5. Calculate the study's power and include a survival curve for the patients.
  6. Add a ROC curve.
  7. Expand the discussions, incorporating more interdisciplinary information.
  8. The conclusions are too short; they should be more comprehensive.

Author Response

  • The introduction is too brief. Provide more information from the literature, explaining in detail what nephrectomy (both total and partial) involves. Also, discuss how modern surgical techniques, including the use of ultrasound, have reached new limits – 10.3390/diagnostics14090942.

Thank you for your thoughtful feedback. We appreciate your suggestion regarding the introduction. In response, we have expanded the introduction to provide a more comprehensive background on nephrectomy. We now include a detailed overview of both radical and partial nephrectomy, highlighting their respective indications, advantages, and limitations. Furthermore, we discuss the evolution of surgical techniques in nephron-sparing surgery, with particular attention to how advancements such as robotic assistance and intraoperative ultrasound have enhanced tumor localization and resection precision. We have also cited recent literature, including the article [Diagnostics 2024, 14(9), 942; https://doi.org/10.3390/diagnostics14090942], to emphasize how these technologies have pushed the boundaries of what is surgically achievable in complex cases such as totally endophytic renal tumors. We believe these additions strengthen the clinical context and rationale for our study.

  • Add a flowchart in Chapter 2.

Thank you for your valuable suggestion. In response, we have added a flowchart in Chapter 2 to visually outline the patient selection process and study design. The flowchart illustrates the overall cohort from the RECORD2 database, the inclusion criteria applied to identify patients with entirely endophytic renal tumors, and the subsequent stratification by surgical approach (open, laparoscopic, robotic). This visual aid enhances the clarity of our methodology and allows readers to better understand the study structure at a glance.

  • Provide more detailed information about the exclusion criteria.

Thank you for your insightful comment. We have revised the “Materials and Methods” section to include a more detailed description of the exclusion criteria. Specifically, we excluded patients who underwent radical nephrectomy, those with exophytic or partially exophytic tumors, incomplete clinical or follow-up data, or missing imaging required to confirm the entirely endophytic nature of the tumor. Additionally, patients with hereditary syndromes, or prior renal surgery on the same kidney were also excluded to maintain a homogeneous study population.

  • Mark statistical significances clearly.

Thank you for your helpful comment. We have updated all relevant tables and figures to highlight statistically significant p-values in bold, making them more visually distinguishable. This adjustment improves clarity and ensures that key findings are easily identifiable for readers.

  • Calculate the study's power and include a survival curve for the patients.

Thank you for your observation. We would like to clarify that while oncologic outcomes such as recurrence-free survival were reported, the study was not specifically powered to detect small differences in oncologic outcomes between surgical approaches. The main focus of our analysis was on perioperative and functional outcomes—particularly renal function preservation—following partial nephrectomy for entirely endophytic renal tumors. Nevertheless, we have included recurrence-free survival data and a Kaplan–Meier curve to provide a comprehensive overview.

  • Add a ROC curve.

Thank you for the suggestion. In response, we have added a ROC curve to evaluate the predictive accuracy of selected variables for >25% renal function loss at 1-month follow-up and Trifecta failure. This analysis provides a visual and quantitative assessment of model performance, supplementing our multivariate analysis. The ROC curve has been incorporated into the results section, along with the corresponding area under the curve (AUC) value, to better illustrate the model’s discriminative ability.

  • Expand the discussions, incorporating more interdisciplinary information.

Thank you for your valuable suggestion. In response, we have expanded the Discussion section to incorporate more interdisciplinary insights. 

  • The conclusions are too short; they should be more comprehensive.

Thank you for your observation. In response, we have revised and expanded the Conclusions section to provide a more comprehensive summary of our findings. The updated section now emphasizes the key perioperative, functional, and oncologic outcomes, highlights the advantages of the robotic approach in terms of early renal function preservation, and reinforces the importance of patient selection and surgical expertise. We also discuss the broader clinical implications and the need for further prospective, randomized studies to validate our results. These additions aim to offer a more complete and impactful closing to the manuscript.